# Synthesis and Characterization of LSX Zeolite/AC Composite from Elutrilithe

**DOI:** 10.3390/ma13163469

**Published:** 2020-08-06

**Authors:** Cailong Xue, Xiaoqin Wei, Zhengwei Zhang, Yang Bai, Mengxue Li, Yongqiang Chen

**Affiliations:** College of Chemistry and Chemical Engineering, Jinzhong University, Jinzhong 030619, China; xuecl@jzxy.edu.cn (C.X.); weixiaoqin@jzxy.edu.cn (X.W.); zzw_dxj@sohu.com (Z.Z.); baiyang@jzxy.edu.cn (Y.B.); limx@jzxy.edu.cn (M.L.)

**Keywords:** hydrothermal synthesis, alkalinity, LSX zeolite/AC composite, gas adsorption, wastewater treatment

## Abstract

The porous carbonaceous precursor obtained from elutrilithe by adding pitch powder and solid SiO_2_ was employed for the first time in an in situ hydrothermal synthesis of LSX zeolite/AC composite. The synthesized samples were characterized by XRD, SEM, and N_2_ adsorption–desorption. The optimum conditions for the hydrothermal synthesis process were set as follows: gelling, aging, and crystallization. The time and temperature required for these steps were 24 h and 65 °C, 12 h and 20 °C, and 48 h and 65 °C, respectively. The molar ratios were (Na_2_O + K_2_O)/Al_3_O_2_ = 7.7, K_2_O/(K_2_O + Na_2_O) = 3. The potential applicability test of the product showed high CO_2_ working capacity, excellent CO_2_/CH_4_ and CO_2_/N_2_ selectivity, and high phenol adsorption capacity. These results suggest that the resultant product has excellent potential value in industrial application.

## 1. Introduction

Elutrilithe, a solid waste product of coal mines, is a kaolinite-rich gangue that is mainly composed of aluminosilicate and carbon. Thus, it is an important source material for the synthesis of porous materials, such as activated carbon (AC) and zeolites. In general, activated carbon from carbonaceous waste material is prepared through two processes that include carbonization followed by activation, which involves both chemical and physical activation [1,2,3]. By following the conventional hydrothermal treatment in alkali solution, the silica and aluminum components of the solid-waste raw materials can be applied to synthesize zeolites, such as analcime, Na-A zeolite, Na-X zeolite, and Na-P zeolite [4,5,6]. However, the previous publications used the carbon content of elutrilithe, and the latter only utilizes the aluminosilicate content, both of them being resource-wasting methods. Therefore, in order to make full use of the main chemical components of elutrilithe, a combination method that includes the chemical activation of starting materials followed by the hydrothermal treatment in alkali solution has been proposed to produce double-function porous materials such as zeolites/activated carbon composites [7] from elutrilithe. In our previous attempts, the effect of the parameters of the thermal treatment (temperature and time) on the activation process has been investigated in detail, and the effects of synthesis conditions for Na-A and Na-X zeolites on the porous activated carbon, a non-traditional precursor in zeolite synthesis, have been reported [8,9].

In recent years, X-zeolite and A-zeolite with unique surface properties and pore structures have been applied in several industrial fields involving catalysts, ion-exchange materials, and adsorbents. Typical X-zeolite is a kind of crystalline aluminosilicate microporous material, having a framework-type code of faujasite (FAU) and a Si/Al molar ratio of nearly 1.25. X-zeolite, due to its prominent characteristics of highly developed porosity and large specific surface area, has proved to be an excellent material for gas separation and purification. A-zeolite, in comparison to X-zeolite, presents a lower porosity and specific surface area, but it has the Si/Al molar ratio of 1. In other words, A-zeolite has more significant negative charges in its skeleton than other types of zeolites, which results in its enhanced ion-exchange capacity and plays an important role in the adsorption and separation of metal ions from wastewater. The ion-exchange capacity of A-zeolite is several times higher than that of X-zeolite. Thus, low silica X (LSX)-zeolite, which contains a skeleton of negative charges similar to A-zeolite and has the porous structure of X-zeolite, has received increasing attention in both preparation and application [10,11,12,13,14,15,16]. As is commonly known, most of the zeolites are prepared by hydrothermal treatment methods as a metastable phase. Therefore, the variables of in the conditions for the preparation of zeolite must be strictly controlled in order to avoid the precipitation of other unwanted but more stable types of zeolites. Some studies have been reported on the preparation of LSX-zeolite by using kaolin [10,11], sodium aluminate, and sodium silicate [12]. However, the conditions for the transformation of AC which is prepared by the carbonization and activation of the porous carbon-based precursor elutrilithe, to LSX-zeolite/AC, have not been reported yet to the best of our knowledge.

In our previous work [7,8,9], many experiments were designed to explore the effects of preparation conditions on the porosity of activated carbon. The preparation procedure obtained from the previous work and the optimized parameters from experimental results were used herein. The effects of activation temperature and time on the porosity of resulting activated carbon were determined. The dried extrudates consisting of elutrilithe, pitch, and SiO2 were calcinated under the flow of N2 and then activated at 850 °C for 24 h under CO_2_ atmosphere. These conditions are considered as standard in the preparation of AC precursor from elutrilithe. Subsequently, the AC precursor was converted into zeolite by hydrothermal synthesis. In the present work, only the effects of hydrothermal treatment and the alkali level on the synthesis of LSX zeolite/AC from porous carbon-based precursor were studied in detail, and further work on the application-oriented properties of the LSX zeolite/AC was performed.

## 2. Experimental

### 2.1. Materials

Elutrilithe, with a major chemical composition of 41 wt % SiO_2_, 35.5 wt % Al_2_O_3_, and 7 wt % C, was obtained from Yangquan in China. High-temperature coal-tar pitch powder with a softening point of 150 °C, coking value of 58.2%, and particle size about 120 mesh, was purchased from Zhenjiang in China. Precipitated silica (ca. 93% SiO_2_) was supplied by a local company (Tong De Chemical Industry Co., Ltd.) in Shanxi, China.

### 2.2. Preparation

The proportion of pitch in the raw mixture was 25%, and the SiO_2_/Al_2_O_3_ molar ratio in the starting mixture was 2.2. Elutrilithe, pitch powder, and SiO_2_ were mixed to produce the AC precursor. The detailed procedure was taken from a previous report [9]. The hydrothermal crystallization of the AC precursor was carried out in a 250 mL round-bottomed flask with a stirring and cooling system. The flask was heated in an oil bath equipped with an automatic temperature controller. The process consisting of three steps (gel formation, aging, and crystallization) was carried out. After the completion of crystallization, the sample was removed from the reactor, filtered, and washed with distilled water until the pH of the filtrate reached 6–7.

A-zeolite [8] and X-zeolite [9] were successfully prepared by combining identical processes that involved calcination and activation under CO_2_ atmosphere followed by a different hydrothermal treatment in alkaline aqueous solution. During this procedure, pitch powder and precipitated silica, which were used to tune the component ratio of the activated carbon in the composites and the Si/Al ratio of zeolite respectively, were added to elutrilithe as starting materials. According to a similar preparation procedure, a porous carbon-based precursor was prepared. Thereafter, the effects of hydrothermal treatment consisting of gel formation, aging, crystallization, and alkali level containing total alkali content, as well as the Na/K ratio on the preparation of LSX-zeolite/AC were investigated. After preliminary trials, the rough range of the synthesis conditions of LSX-zeolite/AC were studied as follows:
Gel formation: temperature (T_1_) 55–75 °C, time (t_1_) 6–48 hAging: temperature (T_2_) 20 °C, time (t_2_) 0–48 hCrystallization: temperature (T_3_) 40–80 °C, time (t_3_) 0–96 hR_1_ ((Na_2_O + K_2_O)/Al_2_O_3_ Ratio): 6.1–8.4R_2_ (K_2_O/(K_2_O + Na_2_O) Ratio): 0.25–0.32


### 2.3. Characterization

Powder X-ray diffraction (XRD, LabX XRD-6000, Kyoto, Japan) with Cu Kα radiation was carried out to determine the mineralogy and crystallinity of the solid products. The Si/Al molar ratios in the samples were determined by XRD [17,18] using the Breck and Flanigen equation: ((192 × 0.00868)/(a_0_-24.191))-1, in which a_0_ is the lattice parameter calculated by XRD. The crystallinities of the samples were determined from the intensity of the XRD characteristic peaks [19,20], and the values were compared to those of commercial X-zeolite and A-zeolite (purchased from China Petroleum and Chemical Co. Ltd, Liaoning, China), which were considered as having 100% crystallinity. The crystallinities of FAU and LTA in the samples were denoted as Xx and Xa, respectively. Thermogravimetric analysis was performed under the flow of oxygen to determine the content of carbon in the composites. Scanning electron microscopy (SEM, Hitachi S4800, Hitachi Ltd., Tokyo, Japan) was employed to observe the crystal size and morphology. N_2_ adsorption–desorption isotherms were measured using an automatic adsorption instrument (1200e, Quantachrome, Inc., Boynton Beach, FL, USA) at the liquid N_2_ temperature (77 K). The samples were all outgassed at 300 °C for 6 h before measurement. The total pore volume (V_T_) was evaluated at a relative pressure of 0.95 and the BET (Brunauer-Emmett-Teller) specific surface area (S_BET_) was calculated from the adsorption branches in the relative pressure range of 0.04–0.25. Micropore volume (V_mic_) and micropore surface area (S_mic_) were calculated by using a t-plot method. The external surface area (S_ext_) was calculated by the formula: S_ext_ = S_BET_−S_mic_.

### 2.4. Adsorption Isotherms of Gases

After drying the samples at 350 °C for at least 4 h, the single-gas adsorption isotherms of CO_2_, CH_4_, and N_2_ were measured at 298 K under both ambient and elevated pressures using a static volumetric system (1200e, Quantachrome Inc., Boynton Beach, FL, USA). The ideal separation factor (ISF) of CO_2_ over CH_4_ or N_2_ was calculated as the ratio of the molar adsorption amount of CO_2_ to that of CH_4_ or N_2_ measured at the same pressure and temperature.

### 2.5. Phenol Adsorption Isotherms

The adsorption isotherms of phenol were obtained at 298 K. For this set of experiments, the mixtures of phenol solutions (initial concentrations (C_0_) of 100 mg/L) and samples (6 g/L) were shaken at 150 rpm for 20 h in a temperature-controlled shaker to ensure equilibrium. The pH of the mixture was adjusted to 6.5 by the addition of HCl (0.1 mol/L) or NaOH (0.1 mol/L). Then, the samples were filtered and the residual phenol concentration was analyzed by the 4-aminoantipyrene method using UV-vis spectroscopy (UV-9600, Beijing, China) at 510 nm. The equilibrium adsorption amount of phenol (q_e_) was calculated by using the equation,
$$q_{e}=\frac{\left(C_{0}-C_{e}\right)\cdot V}{m}$$
where C_0_ and C_e_ are the initial and equilibrium concentrations of phenol (mg/L), respectively. V is the volume of the solution (L), and m is the amount of the adsorbent (g).

## 3. Results and Discussion

### 3.1. Preparation of LSX/AC Composite

#### 3.1.1. Gel Formation

It is an essential to ensure that the porous carbonaceous precursor comes thoroughly in contact with aqueous alkali. In this section, the effects of temperature (T_1_) and time (t_1_) were investigated, while keeping all the other variables constant.

As shown in Figure 1a, the crystallinity of FAU (X_x_) in the results was lower at t_1_ = 6. It increased during the initial gel process and diminished slightly after t_1_ reached 24 h, while the crystallinity of LTA (X_a_) in the composite decreased gradually with the extension of gelling time. The results suggest that the shorter t_1_ did not favor the formation of FAU, which is caused by the incomplete contact between the Si and Al present in the precursors and aqueous alkali. However, too long t_1_ is not suitable for industrial purposes. Thus, the t_1_ = 24 h was selected for this research. Figure 1b displays that the kind of crystal structure obtained from the results was sensitive to the gel temperature. When the T_1_ = 55 °C, the FAU yield only reached 17.4%. The maximum FAU yield was obtained at the T_1_ = 65 °C, and a continuous increase of the T_1_ to 75 °C caused the FAU yield to diminish rapidly. Therefore, the optimum gel temperature was found to be 65 °C.

#### 3.1.2. Aging

In this section, the effect of aging was studied. Some authors have reported that the optimum temperature T_2_ is 20 °C in the synthesis of X-zeolite from various raw materials [21,22,23]. Considering greater demands by the equipment at higher temperatures, the normal laboratory temperature (T_2_ = 20 °C) was selected in this experiment.

Figure 2 shows the effect of t_2_ in X-zeolite synthesis. The X_x_ value increased gradually with the extension of t_2_, while X_a_ showed the opposite tendency. Without the aging step (t_2_ = 0 h), the X_x_ became half of that at an aging time of 12 h. This fact can be explained by the directly proportional relationship between the aging time and nucleation process in the synthesis of X-zeolite. In the event of a too-short aging time, the LTA (Linde Type A) nucleation as the process of generation of an undesired competing phase is facile enough at the present ratio of reactants. Thus, it hinders the crystallization of X-zeolite, which is an observation consistent with previous research [24]. Both the highest X_x_ and the lowest X_a_ occurred at an aging time of 48 h, and it is easy to conclude that the higher X_x_ and the lower X_a_ will be obtained at an aging time above 48 h. However, the variations in both X_x_ and X_a_ become smaller as t_2_ increases to above 12 h. Therefore, given the increased value of X_x_, the extension of aging time cannot compensate for the cost of equipment required in the industrial production of zeolite. In this case, the aging time was determined to be 12 h.

#### 3.1.3. Crystallization

Crystallization mostly contributes to the growth of the crystal nucleus formed during aging into zeolite crystals. Even a slight change in the crystallization conditions may produce very different zeolite phases. Thus, to obtain the desired type of zeolite by using a porous carbonaceous precursor, the effects of temperature (T_3_) and time (t_3_) of crystallization on the X-zeolite were investigated.

In Figure 3, the values X_x_ and X_a_ are plotted as a function of T_3_ (Figure 3a) and t_3_ (Figure 3b), respectively. T_3_ is the most important variable in the crystallization process, because even a slight change in T_3_ can result in different products (Figure 3a). At first, the synthesis was performed at T_3_ = 50 °C, and the result shows little crystalline product. The major product was an almost amorphous phase, along with small amounts of FAU. When T_3_ < 65 °C, the values of X_x_ were found to be low in all the samples, which can be explained by considering the relationship between the energy of crystal growth and the crystallization temperature. If the temperature is too low, the crystal growth is very slow, or the system can proceed even without crystal growth, since a lower temperature is insufficient to provide the energy required for the crystal growth. When T_3_ > 65 °C, the X_x_ values dropped rapidly, and when T_3_ was set to 80 °C, the X_x_ value was less than the half of that at 65 °C. Consequently, the optimum value of T_3_ was 65 °C. Figure 3b clearly shows that the X_x_ value exhibited a significant increase with the increment in t_3_ before t_3_ = 48 h, and then, it displayed a slight decrease in the trend. Accordingly, it can be observed that the optimum t_3_ value is 48 h.

It is interesting to note that the optimum value of T_3_ (Figure 3a) in this experiment is lower than that in other studies [10,12]. It is also different from the result reported by Dalai [25], wherein the X-zeolite starts crystallizing at higher temperatures from an initial mix containing less SiO_2_ and more Al_2_O_3_. These results can be attributed to the special reactants, namely the porous carbon-based precursors without considering the solubility of silicate ions. Besides, previous research in this field reported that X-zeolite needs a longer time than A-zeolite to be fully formed [26], because FAU is more complex and has larger polymeric silicate units than LTA. Yet, Figure 3b clearly shows that the longer t_3_ resulted in a higher X_a_ value up to t_3_ = 48 h and a lower X_x_ thereafter. By combining the changing trend of X_a_ with T_3_ (Figure 3a), it can be concluded that the higher temperature and longer crystallization time are favorable for the formation of a more stable LTA structure under the existing conditions. This view is also supported by the findings of our previous work [8].

### 3.2. The Alkali Levels

#### 3.2.1. R_1_ ((Na_2_O + K_2_O)/Al_2_O_3_ Ratio)

The influence of alkalinity on the properties of the products was studied by changing the number of moles of Na_2_O + K_2_O in the hydrothermal treatment process. The variable R_1_ was analyzed at the values of 6.1, 6.9, 7.7, and 8.4, respectively, by adding sodium hydroxide and potassium hydroxide, when the other parameters were kept constant, and the results are shown in Figure 4. For R_1_ < 7.7, the values of X_x_ and X_a_ increased with the increase in of alkalinity. For R_1_ > 7.7, the values of X_a_ were observed to decrease significantly, but a further increase in alkalinity resulted in the lowering of X_x_ values at the current reaction time. The reason could be that the framework of zeolite got dissolved in aqueous alkali, which has very strong basicity when R_1_ > 7.7. Therefore, the R_1_ = 7.7 was selected. By analyzing the value of R_1_ reported by others [10,11,12], it can be stated that the optimum value of R_1_ in this experiment is obviously greater than those from others. This phenomenon can be attributed to two factors. First, unlike other raw materials dissolved in aqueous alkali, the columnar activated carbon precursor with a large number of channel structures comes into full contact with aqueous alkali by maceration. Thus, it could be more difficult to obtain good contact with aqueous alkali at low concentrations, leading to higher alkali concentration requirements. In addition, since the activated carbon is hydrophobic, it is less prone to contact between the precursor and aqueous alkali.

#### 3.2.2. R_2_ (K_2_O/(K_2_O + Na_2_O) Ratio)

Our previous work has verified that A-zeolite was obtained at certain experimental conditions without adding potassium hydroxide in aqueous alkali [8]. Figure 5 shows the effect of R_2_ on the properties of the resultant materials. The products displayed a rapid transformation from A-zeolite to X-zeolite with increasing R_2_. No further increase occurs for X_x_ after R_2_ reaches 0.30, while X_a_ continues to drop with increasing R_2_. Thus, it can be concluded that pure X-zeolite can be obtained if R_2_ increases further. Given the price of potassium hydroxide and the purity of zeolite required for industrial purposes, R_2_ = 0.3 was selected. The above results confirmed that potassium played a key role in the synthesis of LSX. The constant proportion of LSX and A-zeolite in the products has a strong dependency on the K_2_O/(K_2_O + Na_2_O) ratio. By varying the K_2_O/(K_2_O + Na_2_O) ratio, the products with different constant proportions of LSX and A-zeolite can be directly prepared according to the requirements to have a potential application value.

### 3.3. Characterization

The main properties of LSX-zeolite/AC composite synthesized at optimum conditions as described above (T_1_ = 65 °C, t_1_ = 24 h, T_2_ = 20 °C, t_2_ = 12 h, T_3_ = 65 °C, t_3_ = 48 h, R_1_ = 7.7, R_2_ = 0.3) were studied. As shown in Figure 6, the characteristic peaks of zeolite with both FAU and slight LTA structures were observed from the XRD patterns of the zeolite/AC sample. The FAU (53.5%) and LTA (5.2%) results suggest that the zeolite phase of the composite is predominantly composed of X-zeolite. Figure 7 is the SEM image of the sample, wherein the amorphous particles are observed but the special octahedral crystal is not seen, indicating that the crystal aggregates are surrounded by activated carbon. The X-zeolite with the Si/Al ratio lower than 1.15 is often called LSX [27]. Combining the values of the Si/Al ratio from Table 1 with the above results, it can be concluded that the LSX-zeolite/AC composite was obtained under the synthesis condition described above.

The average particle size of the LSX-zeolite/AC composite shown in Figure 7 is approximately 8 μm, which is higher than that of the standard LSX-zeolite (about 3 μm). This significant difference is likely related to the raw materials because the AC precursor is different from other silicon and aluminum sources. On the other hand, the high agglomeration and intergrowth between LSX and AC as observed in the SEM image (Figure 7) can be a possible reason for the relatively large average particle size.

In addition, the cauterization loss of sample is 15.6%, and the total crystallinity of the sample is 58.7%, which corresponds to a small amount of unconverted reactant that is observed from the XRD pattern in Figure 6 and SEM image in Figure 7.

Figure 8 shows the N_2_ adsorption–desorption isotherms of the LSX/AC LSX at 77 K. The standard LSX exhibits a type-I isotherm according to IUPAC (International Union of Pure and Applied Chemistry) classification, suggesting its exclusively microporous structure. The LSX-zeolite/AC samples exhibit intermediates between type I and IV isotherms, which indicate the presence of both microporous and mesoporous structures [28]. This observation confirms that the LSX-zeolite/AC composite was successfully obtained. The pore structure parameters of the LSX/AC and LSX by N_2_ adsorption–desorption isotherms at 77 K are listed in Table 1. The specific surface area of the sample was found to be 465 m^2^/g, which is lower than that for the standard LSX. This result can be attributed to the activated carbon with fewer pores compared to zeolite.

### 3.4. Gas Separation

Adsorption equilibrium isotherms of N_2_, CH_4_, and CO_2_ adsorbed on the LSX/AC sample at 298 K in the 0–100 kPa pressure range are presented in Figure 9a. Apparently, all three gases show limited adsorption capacity on LSX-zeolite/AC compared to their adsorption capacity on the absorbents of 13X-, 5A- and NaY-zeolite under conditions similar to those mentioned in Table 2. This is due to the introduction of activated carbon with a fewer number of pores than on LSX/AC-zeolite. Figure 9b shows the tendency of variation of ideal separation factors (ISF) of CO_2_/CH_4_ and CO_2_/N_2_ on LSX-zeolite/AC at 298 K under the 0–100 kPa pressure range. Obviously, both selectivities decrease with the increase in the adsorption pressure. The ISF for CO_2_/CH_4_ and CO_2_/N_2_ are 28 and 80 at 10 kPa and decrease to 7 and 18 at 100 kPa. Compared with the two ISF values of 13X-, 5A-, and NaY-zeolites in Table 2, both the ISF values of LSX-zeolite/AC are similar to those of the 13X- and NaY-zeolites, and higher than that of the 5A-zeolite. Working capacity is a standard critical parameter for evaluating the adsorbents in practical industrial applications. It is defined as the difference between the adsorbed amounts at adsorption and desorption conditions [29]. From pure CO_2_ isotherm, the CO_2_ working capacities between 100 and 10 kPa for LSX/AC-zeolite were calculated. Remarkably, as shown in Table 2, the CO_2_ working capacity (38.4 cm^3^·g^−1^) obtained in the present work is much greater than the reported values for the 13X-, 5A-, NaY-, and LSX-zeolites. Thus, the LSX-zeolite/AC is a good candidate for CO_2_ capture and sequestration from flue gas due to its high CO_2_/CH_4_ and CO_2_/N_2_ selectivity.

### 3.5. Wastewater Treatment

Phenols, among the key pollutants in wastewater, are highly toxic even at low concentrations and difficult to be biodegraded [35]. Therefore, it is essential to remove phenols from wastewater before they are released to the environment. The phenol adsorption isotherms of LSX, 5A, and LSX/AC adsorbents are shown in Figure 10. Interestingly, the phenol adsorption capacity and adsorption rates at the initial stages on LSX-zeolite/AC are much higher than those on LSX- and 5A-zeolites, although the LSX-zeolite/AC displays lower specific surface area than LSX-zeolite. This result can be contributed to the accessibility of the adsorption sites, which are enhanced with a larger external specific surface area and pore volume when activated carbon is compounded into the zeolite. These excellent properties of LSX-zeolite/AC adsorbent make it a promising adsorbent for the removal of phenols from aqueous solutions.

## 4. Conclusions

For the first time, the synthesis of LSX-zeolite/AC by a hydrothermal treatment was successfully accomplished through the AC precursor from elutrilithe. The best product was obtained by using the following parameter values: t_1_ = 24 h and T_1_ = 65 °C, t_2_ = 12 h and T_2_ = 20 °C, t_3_ = 48 h and T_3_ = 65 °C, R_1_ = 7.7 and R_2_ = 3.0, respectively. The product with FAU crystallinity of 53.5% and a cauterization loss of 15.6% exhibits a diameter of about 8 μm, a specific surface area of 465 m^2^/g, and a framework Si/Al of 1.06. The gas adsorption measurements indicate that the product has superior CO_2_/CH_4_ and CO_2_/N_2_ selectivity, especially under low pressures compared to previous reports. The CO_2_ working capacity is higher than the values reported for 13X-, NaY-, and 5A- zeolites. In addition, the LSX/AC adsorbent exhibits excellent adsorption performance for phenol. The simple synthesis method and the low investment cost makes the proposed synthesis of LSX-zeolite/AC suitable for easy scale-up. Thus, the resultant products would have excellent potential value in industrial applications.

## Figures and Tables

**Figure 1 materials-13-03469-f001:**
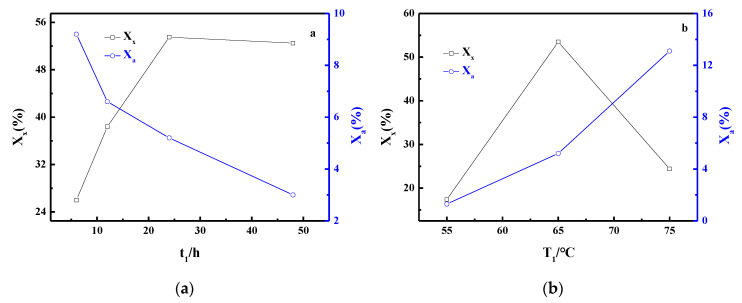
Influence of t_1_ (**a**) and T_1_ (**b**) on crystallinities of zeolite.

**Figure 2 materials-13-03469-f002:**
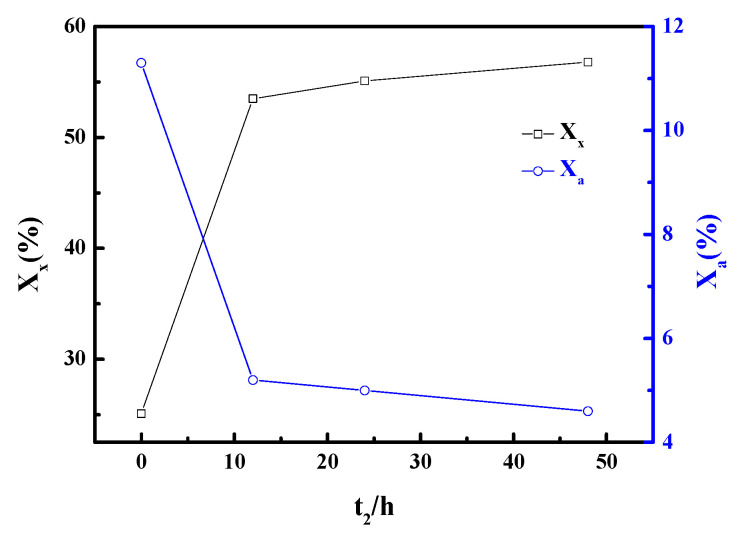
Influence of t_2_ on crystallinities of zeolite.

**Figure 3 materials-13-03469-f003:**
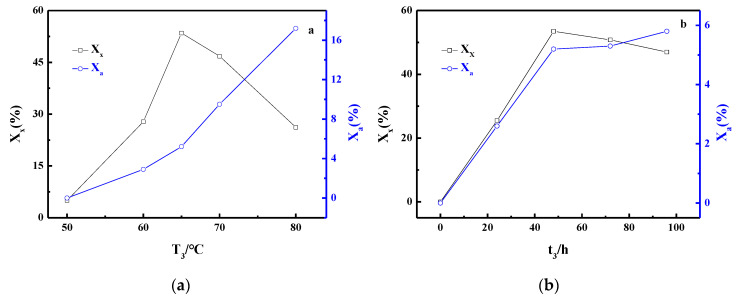
Influence of T_3_ (**a**) and t_3_ (**b**) on crystallinity of zeolite.

**Figure 4 materials-13-03469-f004:**
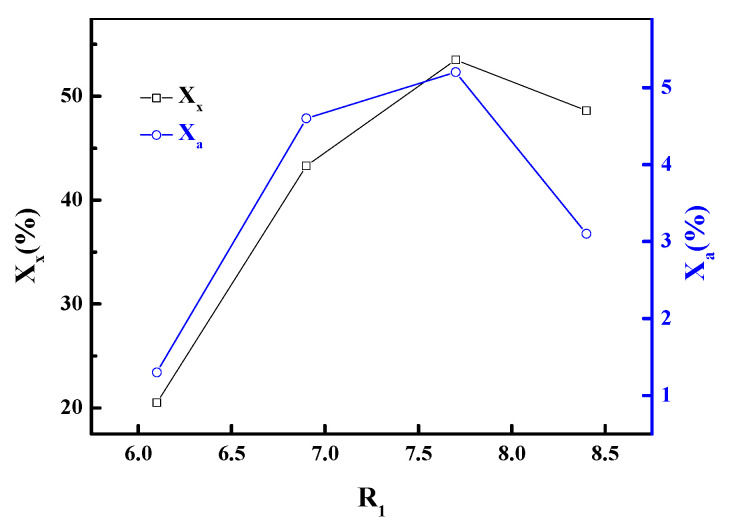
Influence of R_1_ on crystallinities of zeolite.

**Figure 5 materials-13-03469-f005:**
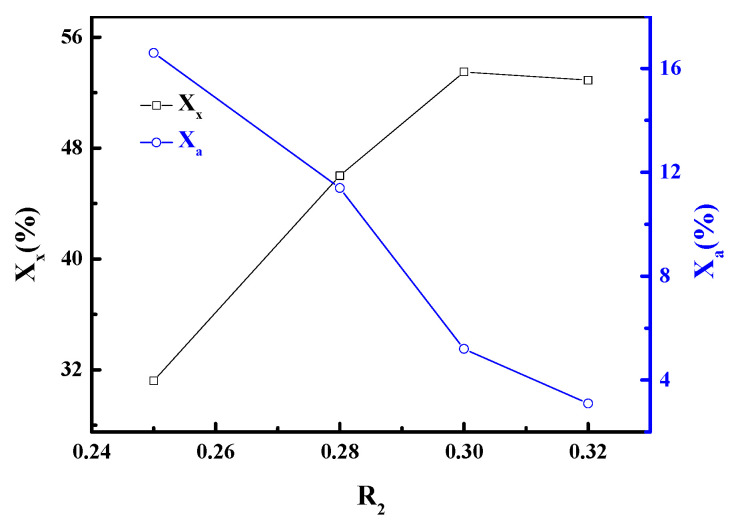
Influence of R_2_ on crystallinities of zeolite.

**Figure 6 materials-13-03469-f006:**
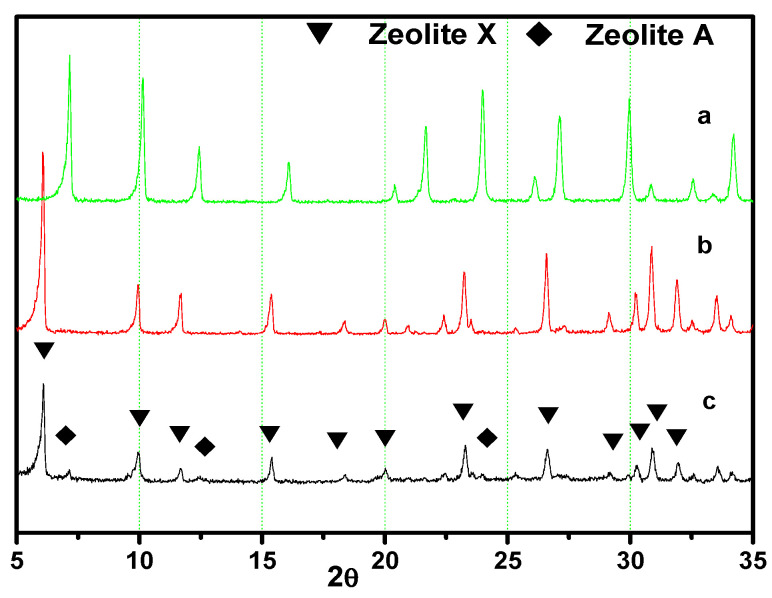
XRD patterns of A-zeolite (**a**), X-zeolite (**b**), and low silica X (LSX)-zeolite/activated carbon (AC) composite (**c**).

**Figure 7 materials-13-03469-f007:**
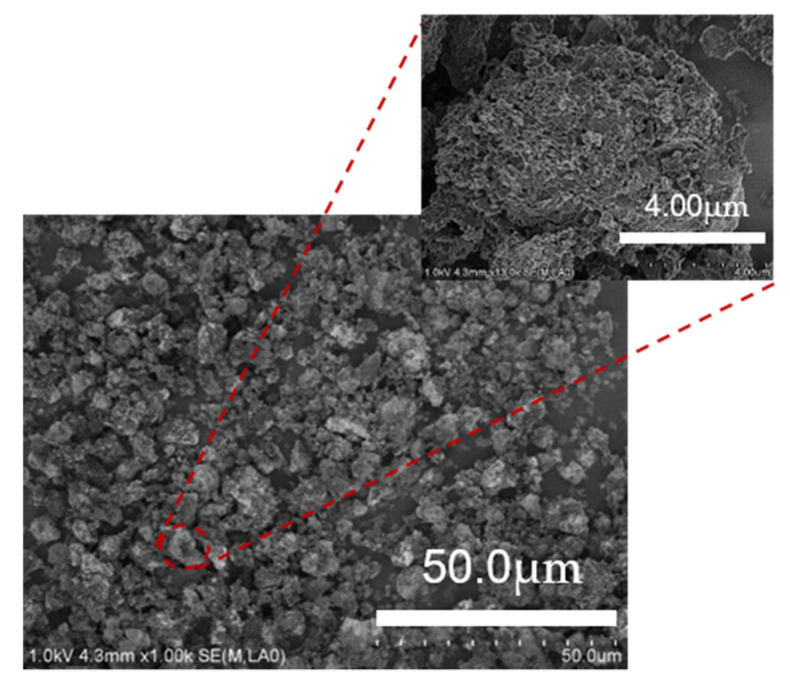
SEM images of synthesized LSX-zeolite/AC.

**Figure 8 materials-13-03469-f008:**
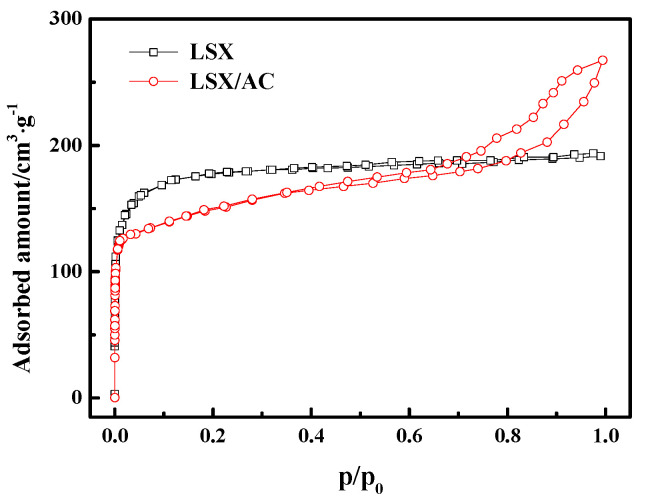
N_2_ adsorption–desorption isotherms of the LSX/AC and LSX at 77 K.

**Figure 9 materials-13-03469-f009:**
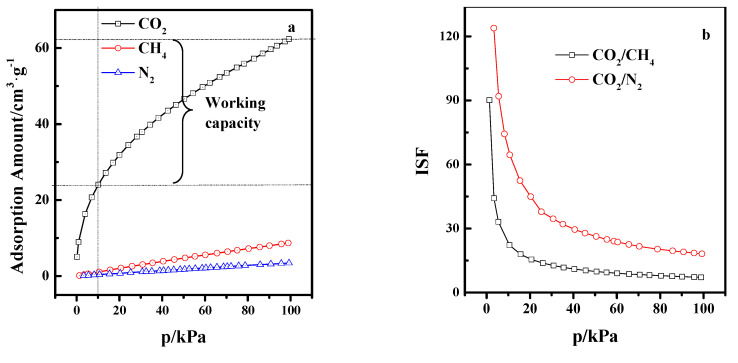
CO_2_, CH_4_, and N_2_ adsorption isotherms on LSX-zeolite/AC at 77 K (**a**), and ISFof CO_2_/CH_4_ and CO_2_/N_2_ (**b**).

**Figure 10 materials-13-03469-f010:**
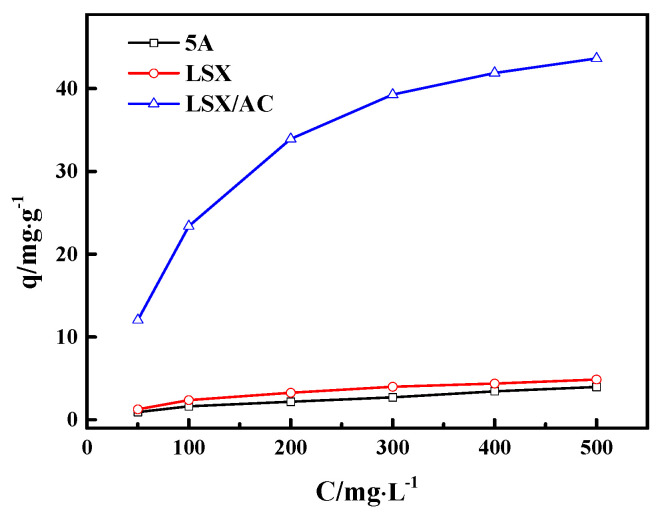
Adsorption isotherms of phenol on samples.

**Table 1 materials-13-03469-t001:** Pore structure parameters and Si/Al ratio of the samples.

Samples	S_BET_ (m^2^/g)	S_mic_ (m^2^/g)	S_ext_ (m^2^/g)	V_mic_ (cm^3^/g)	V_total_ (cm^3^/g)	Si/Al
LSX/AC	465	340	125	0.174	0.407	1.06
LSX	532	507	25	0.265	0.300	1

**Table 2 materials-13-03469-t002:** CO_2_, CH_4_, and N_2_ adsorption properties of various samples. ISF: ideal separation factor.

Adsorbent	Adsorption Capacity (cm^3^·g^−1^)			ISF at 100 kPa		Working Capacity (cm^3^·g^−1^)	References
	CO_2_	CH_4_	N_2_	CO_2_/CH_4_	CO_2_/N_2_		
13X ^a^	110.0	15.9	6.1	6.9	18.0	34.1	[30]
NaY ^a^	97.8	11.2	6.5	7.3	18.4	35.6	[31,32]
5A ^b^	115.6	20.6	13.5	5.6	8.5	32.8	[33]
LSX ^c^	99.3	-	-	-	-	30.5	[34]
LSX/AC ^b^	62.4	8.6	3.4	7.3	18.3	38.4	This work

^a^ 295 K, 100 kPa; ^b^ 298 K, 100 kPa; ^c^ 303 K, 100 kPa.

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
