# Peer review of "Synthesis and Characterization of LSX Zeolite/AC Composite from Elutrilithe"

_materials, 2020, doi:10.3390/ma13163469_

Round 1

Reviewer 1 Report

I carefully analyzed the manuscript and this is the current situation:

1) English has been revised and has now improved significantly

2) but the text (excepting for english corrections) is the same as the first version with minimal changes consisting of:

adding a sentence (line 85-89)

deleting a 9-word phrase (line 242)

modification of fig. 7

adding two references.

3) my previous suggestion for major and minor revision have been ignored or have had insufficient answers.

My final opinion is that the text has improved only for English quality and not at all in presentation and discussion that are unchanged. 

Reviewer 2 Report

Dear Authors, 
The present manuscript deals with a quiet interesting subject regarding LSX zeolite/AC composites using elutrilithe as precursor. The main idea of the manuscript is interesting, in the opinion of the Reviewer, it could be published after addressing some issues. See below some of my concerns regarding the manuscript in the Comments section. Also, I wish all the best for the authors and I hope that the manuscript will be acceptable for publication after revision and improvements. 

Best regards, 
The Reviewer 

0) Some minor typos and grammar errors can be found in manuscript, few examples (mainly from the first page):  
- row 9: in situ should be written nearly in all the cases in italic
- some commas are missing from the first sentence (row 22-24) suggestion: "...gangue, mainly composed of aluminosilicate and carbon, that it makes..."))
- word "synthesis" was used twice in the same sentence (row 23-24), maybe you can remove the second one (similar to row 99, with "...drying the samples were dried...")
- row 35 - effectts.
Also, in some places revision of the language would help the lucidity of the text,e.g.:
- row 35: instead of "...temperature and time" you could use "the effect of parameters of he thermal treatment (temperature and time" 
- row 36: instead of "However, the former only uses..." it would be better to use "However, the previous publications used the..."
- row 36: I suppose "synthesis condition" should be plural (conditionS), as there is probably not just one parameter)
- row 41: please introduce the abbreviation of FAU
...etcetc...
If you have the possibility, I would recommend a professional editing service or ask for a help of native English speaker!

1) I think, that Table 1 should be omitted, as the chemical composition of the untreated elutrilithe can be summarized in half sentence in the main text (for example: The chemical composition of the eluthrilite was determined by..., as follows: 41% of SiO2, 35.5% of Al2O3 and 7% of C).
2) Section 3.1. (without sub-sections) would fit more in the Experimental section, as describes the preparation, no real results or discussion is included in this section!
3) Figure 7 inset is slightly dark, please increase its brightness!
4) At section 3.5. in the maintext you say, that isotherms of 13X, 5A and LSX/AC is presented...but the legend of the fig is about 5A, LSX and LSX/AC...please, revise it!

Reviewer 3 Report

In this report, authors synthesized LSX-zeolite/Activated carbon by hydrothermal method from elutrilithe. They characterized prepared samples with XRD, SEM, and BET technique. They studied effect of synthesis parameters and tested gas and phenol adsorption activity.

1. "The Si/Al molar ratios in the samples were determined by means of chemical analysis and the XRD method." How authors estimated Si/Al molar ratio from XRD method?
Discussion can be included.
2. Absorption capacity of LSX can be compared and included in the Table 3.
3.LSX/AC adsorbent exhibited excellent adsorption performance for phenol even though it has lower surface area than LSX? Why? Authors should explain clearly.
4. Authors should check typos carefully.
5. Page 2, line 57, Authors should include more details for their previous study.
6. Authors should cite following reference Korean Chemical Engineering Research, Vol.45, No.2, 160-165, April, 2007

Round 2

Reviewer 1 Report

English was improved, but I could find only very few improvement in the content of text.

This manuscript is a resubmission of an earlier submission. The following is a list of the peer review reports and author responses from that submission.

Round 1

Reviewer 1 Report

The paper “Synthesis and characterization of LSX zeolite/AC composite from elutrilithe” presents a study on the use of elutrilithe for a new synthesis of LSX-zeolite/AC by hydrothermal treatment. The preparation parameters T, time and composition of alkali solutions, were tested. Some functional tests showed high CO2 working capacity, very good CO2/CH4 and CO2/N2 selectivities and high phenol adsorption capacity of the prepared LSX/AC. Authors suggest that the proposed synthesis may be a simple synthesis method of LSX-zeolite/AC that could be easily scaled-up. The paper presents a good novelty, but it needs some improvements:

Major revisions:

1) The paper analyzes the experimental conditions of the preparation, but in my opinion, there might be a problem about the reproducibility of these experiments. Authors should verify if the results are reproducible, and/or in any case discuss what they know about that aspect.

2) Figure 1 :  It is not clear what exactly is Xx and Xa= crystallinity of the composite. It should be defined in experimental.

3) For a composite: how are crystallinities measured? how are they calculated? Authors should add more details about how authors discriminated the crystallinity of the two components of the composite.

4) Figure 7. The two images have to be better described and commented. What are they referred to? What do they show exactly? Which are the relevant differences of the two images?

5) Authors have to better define the “working capacity” and how it is determined in Figure 9.

Minor revisions:

1) The caption of all figures CONSIST ONLY IN A TITLE. That is good, but not sufficient. All the captions shoud be integrated with more details about the samples or data reported, the experimental conditions and the units on axis.

2) English need a deep revision: some examples…

Line 14: application test of produce

Line 53: some investigates

Line 62: were investigated

Line 248: adsoptiom epuilibrium

Line 282: the product show

Reviewer 2 Report

Comments to Author

I can not see any novelty from this study.

The study is about the synthesis and characterization of conventional composite materials via hydrothermal or sol gel method, both of which are nothing new to report. Plenty of articles have been published in the past decades, with a versatile of advanced approaches and characterization techniques.

Secondly, the manuscript is not well organized in a logical approach, with proper experimental findings and statements. Meanwhile, the language is not well polished with obvious mistakes and grammar errors. As a open access journal, materials should have the baseline to accept high quality manuscript which can be readable and clear without misunderstanding.

Thirdly, the application for wastewater treatment is quite simple, and can not reveal any significant points to the readers. A substance enhancement should be conducted as well.

Lastly, the figures should be well plotted and edited with clear images and proper errors bars to show its accuracy and STD.